# Phytochemicals in Skeletal Muscle Health: Effects of Curcumin (from *Curcuma longa Linn*) and Sulforaphane (from *Brassicaceae)* on Muscle Function, Recovery and Therapy of Muscle Atrophy

**DOI:** 10.3390/plants11192517

**Published:** 2022-09-26

**Authors:** Nancy Vargas-Mendoza, Eduardo Madrigal-Santillán, Isela Álvarez-González, Eduardo Madrigal-Bujaidar, Liliana Anguiano-Robledo, José Leopoldo Aguilar-Faisal, Mauricio Morales-Martínez, Luis Delgado-Olivares, Elda Victoria Rodríguez-Negrete, Ángel Morales-González, José A. Morales-González

**Affiliations:** 1Laboratorio de Medicina de Conservación, Escuela Superior de Medicina, Instituto Politécnico Nacional, Plan de San Luis y Díaz Mirón, Col. Casco de Santo Tomás, Del. Miguel Hidalgo, Mexico City 11340, Mexico; 2Escuela Nacional de Ciencias Biológicas, Instituto Politécnico Nacional, Unidad Profesional A. López Mateos, Av. Wilfrido Massieu. Col., Zacatenco, Mexico City 07738, Mexico; 3Laboratorio de Farmacología Molecular, Escuela Superior de Medicina, Instituto Politécnico Nacional, Plan de San Luis y Díaz Mirón, Col. Casco de Santo Tomás, Del. Miguel Hidalgo, Mexico City 11340, Mexico; 4Licenciatura en Nutrición, Universidad Intercontinental, Insurgentes Sur 4303, Santa Úrsula Xitla, Alcaldía Tlalpan, Mexico City 14420, Mexico; 5Centro de Investigación Interdisciplinario, Área Académica de Nutrición, Instituto de Ciencias de la Salud, Universidad Autónoma del Estado de Hidalgo, Circuito Actopan-Tilcuauttla, s/n, Ex Hacienda la Concepción, San Agustín Tlaxiaca, Hidalgo 2160, Mexico; 6Servicio de Gastroenterología, Hospital de Especialidades Centro Medico Nacional Siglo 21, Mexico City 06720, Mexico; 7Escuela Superior de Cómputo, Instituto Politécnico Nacional, Av. Juan de Dios Bátiz s/n Esquina Miguel Othón de Mendizabal, Unidad Profesional Adolfo López Mateos, Mexico City 07738, Mexico

**Keywords:** phytochemicals, curcumin, sulforaphane, skeletal muscle

## Abstract

The mobility of the human body depends on, among other things, muscle health, which can be affected by several situations, such as aging, increased oxidative stress, malnutrition, cancer, and the lack or excess of physical exercise, among others. Genetic, metabolic, hormonal, and nutritional factors are intricately involved in maintaining the balance that allows proper muscle function and fiber recovery; therefore, the breakdown of the balance among these elements can trigger muscle atrophy. The study from the nutrigenomic perspective of nutritional factors has drawn wide attention recently; one of these is the use of certain compounds derived from foods and plants known as phytochemicals, to which various biological activities have been described and attributed in terms of benefiting health in many respects. This work addresses the effect that the phytochemicals curcumin from *Curcuma longa* Linn and sulforaphane from *Brassicaceae* species have shown to exert on muscle function, recovery, and the prevention of muscle atrophy, and describes the impact on muscle health in general. In the same manner, there are future perspectives in research on novel compounds as potential agents in the prevention or treatment of medical conditions that affect muscle health.

## 1. Introduction

The mobility of the human body is supported by the skeletal muscle, which is a highly plastic organ that fulfills vital metabolic and endocrine functions and participates in the respiratory process and in the body’s physical posture. The functionality of the skeletal muscle is determined by, among other things, its contractile capacity, in which an extensive network of molecular mechanisms that regulate the signaling pathways responsible for protein synthesis and degradation is intricately involved, promoting the balance between hypertrophic and atrophic signals. Under healthy conditions, an equilibrium between protein synthesis and degradation is maintained, but there are several situations that break the balance, favoring proteolysis in the fibers and producing muscle atrophy [1]. Physiological situations such as aging itself or numerous chronic diseases such as AIDS, cancer, kidney failure, chronic obstructive pulmonary disease, malnutrition, and immobility due to trauma, amputation, and neurodegenerative disease, etc., severely compromise the muscle health of humans. Muscle atrophy is the result of a decreased cross-sectional area (CSA) of muscle fibers, leading to the loss of muscle strength and contractile capacity [1,2]. Fundamentally, the synthesis of muscle mass occurs as a result of the conjunction of elements that involve the activation of signaling pathways and myogenesis. The phosphatidylinositol 3-kinase (PI3K)/serine threonine kinase (Akt) pathway is one of the most important anabolic signaling pathways that stimulates the target of rapamycin (mTOR) in mammals involved in the synthesis of muscle mass. Thus, the PI3K/Akt/mTOR pathway is the main one responsible for myofibrillar protein synthesis and is regulated by the hormones testosterone and insulin, the insulin-like growth factor 1 (IGF-1), leucin, and exercise. The activation of the PI3K/Akt/mTOR pathway will be crucial for enhancing hypertrophy [3]. Myogenesis is triggered by muscle injury or by mechanical stimuli such as exercise, performed by satellite cells or muscle stem cells, which command a myogenic differentiation program that restores muscle tissue by forming new fibers. Contrariwise, muscle atrophy is influenced by proatrophic factors such as inflammation, oxidative stress, and mitochondrial damage; these factors play an important role in protein degradation through the ubiquitin proteasome system (UPS), which coordinates the main protein catabolic pathways upregulating the ubiquitin ligases atrogin-1, muscle RING finger1 (*MuRF-1*), and casitase-B-lineage lymphoma (Cbl-b) [2].

Oxidative stress and inflammation are closely related with chronic diseases and aging; for instance, the elevation of tumor necrosis factor alpha (TNF-α) along with the increase in the proinflammatory cytokines interleukin-6 (IL-6) and interleukin-1β (IL-1β) promote the nuclear factor kappa beta (NF-κB) transcription factor, leading to protein degradation and acting as an inducer of *MuRF-1* and atrogin-1 binding directly to the *MuRF-1* promoter [4]. NF-κB, along with other catabolic factors such as Forkhead box O (FoxO), p50 and p38 mitogen-activated protein kinase (p38-MAPK), enhance the transcription of E3 ubiquitin ligase genes including *MuRF-1* and *FBox* (*MAFbx*/atrogin-1) [5,6,7]. Autophagic-related genes can also be related with the activation of these pathways in protein breakdown by the UPS, such as microtubule-associated protein 1A/1B-light chain 3 (LC3) and the Bcl2/adenovirus E1B 19-kDa-interacting protein 3 (Bnip3) by means of the autophagic-lysosomal pathway [8,9]. Mostly, the impairment of the PI3K/Akt/mTOR pathway activates UPS; frequently, signaling activation initiates with the disruption of the binding of IGF-1 to its tyrosine kinase receptor GF-1R, this binding producing transphosphorylation of the receptor, and the phosphorylated tyrosines generate a docking site for the recruitment of the insulin receptor substrate 1 (IRS1) [10]. The phosphorylation of IRS1 is a key step in the regulation of IGF1 signaling. It has been reported that IRS1 degradation by inducing ubiquitin ligase Cbl-b and the activation of FoxO and FoxO-mediated ubiquitin ligases are also related with elevated oxidative stress and inflammation [11]

Moreover, myogenesis, which is regulated by the presence of myogenic factors such as myogenin, the myoblast determination protein (MyoD), myogenic factor 5 (Myf5), and myogenic regulatory factor 4, is involved in a redox-sensitive mechanism. During muscle regeneration, reactive oxygen species (ROS) and reactive nitrogen species (RNS) are considered crucial factors for satellite-cell biology through their modulating a wide range of the cellular processes involved [12]. However, excessive ROS/RNS might lead to an imbalance between antioxidant defenses and the ability to counteract the harmful effects on the cell, which is a major issue in the impairment of protein synthesis and myogenesis, favoring muscle atrophy. A strong association has been reported of the activation of the two main proteolytic systems, that is, the ubiquitin–proteosome and the lysosomal autophagy pathways, with increased oxidative stress related with protein degradation and age-dependent muscle atrophy [13]

The disruption of the anabolic mechanisms involved in regeneration and myogenesis reduces the capacity for mobility and independence, leading to frailty and poor quality of life; this entails repercussions, not only on the health of individuals, but also on the economic budgets allocated to health services at regional and global levels [14]. Muscle atrophy is a chronic condition characterized by loss of function and structural muscle damage in which there is an imbalance between proteolysis and muscle protein synthesis. Numerous reasons can be the cause of muscle deterioration, including aging, chronic diseases, starvation, sedentary lifestyle, lack of exercise, or overtraining without appropriate rest–recovery period. Although muscle atrophy is identified as a cause of immobility that contributes to frailty and poor quality of life, to date, not enough alternatives have been generated for the prevention or treatment of this condition. In recent years, widespread research has emerged on different therapeutic options that might aid in reducing the risk of developing muscle affectations. Some compounds that are derived from plants or foods, denominated phytochemicals, have attracted attention because their biological activity has been described as antioxidant, cardioprotector, antidiabetic, and anticancer [15]. In agreement with this, numerous in vivo/in vitro studies have found that phytochemicals such as polyphenols have been proven to encourage muscle recovery and could be used for the treatment of muscle atrophy [16]. This review describes in organized form the manner in which the phytochemicals curcumin (*Curcuma longa*) and sulforaphane (SFN) (*Brassicaceae*) have proven to exert an impact on the molecular mechanism signaling involved in muscle function at different levels. A growing body of evidence indicates that both curcumin and SFN trigger signaling intricate in-muscle protein synthesis while inhibiting protein degradation. Thus, they can be considered potential targets of the use of nutraceuticals in the treatment of muscle atrophy and recovery, possessing a positive impact on muscle health.

## 2. Curcumin

### 2.1. Botanical Description of Curcuma longa Linn

The rhizome *Curcuma longa* Linn (*C. longa*) belongs to the Zingiberaceae (ginger) family and is a perennial sterile plant commonly known as turmeric, which is cultivated and widely distributed in Asian nations. India and China have a long history in the use of *C. longa* in their traditional medicine, in cosmetics, and among spices in food. It is known as “The Golden Spice” in India. Turmeric is a tall herb with no stem and rootstock, reaching 1 m in height and flourishing in tropical and subtropical regions at temperatures ranging between 20 and 30 °C, requiring constant rainfall to grow and only reproducing through its rhizomes. The leaves are around 1 m in length, with long dark-green leaves on the upper surface and light-green leaves beneath; these leaves are oblong or lanceolate in shape, and take the appearance of spikes prior to that of leaves. The flowers are pale yellow in color and reddish on the top, while the flowering bract is green and intensely ferruginous purple in color. It has 2-meter-long pseudostems with 8–12 leaves proceeding from these. The ripe rhizomes have a rough and segmented skin; from the inside, they are yellowish-brown in color with a dull orange hue. Small pointed or conical tubers are 2.5–7.0 cm in length and 2.5 cm in diameter, sprouting from the main rhizome [17,18]. *C. longa* optimally grows in a humid environment, and exposure to sunlight facilitates the rhizome in achieving larger and better rhizomes, thus extending harvest times from January to March–April. Turmeric rhizomes require rich soil conditions, proper nutrients, and a small sand content; these grow ideally in irrigated and rain-fed areas in light-black ashen loam soil and in red-to-stiff loam soils. Turmeric is known to have originated in South and Southeast Asia in China, and in western India and Vietnam; it is currently cultivated in various Asian countries such as Nepal, Thailand, Malaysia, Cambodia, Madagascar, the Philippines, Indonesia, Bangladesh, and the state of West Bengal in India, as well as others.

From the grinding of the dry rhizome, a yellow powder is obtained that has a bitter and sweet taste; this powder is employed in traditional cuisine as a spice to provide color, flavor, and aroma, and it is frequently used as a food preservative or additive. In Chinese, Indian, Japanese, and Korean traditional medicine, turmeric has been utilized as a therapy for different affectations, due to its multiple benefits as an anti-inflammatory, anticancer compound, and analgesic; to treat skin disorders and wounds; and as an antidiabetic, liver protector, cardioprotector, neuroprotector, and antimicrobial with antiseptic effects, among many others [19]. These mentioned benefits are attributed to its components called curcuminoids, and curcumin comprises the most abundant and active bioactive compound (77%) (1,7-bis[4-hydroxy-3-methoxyphenyl]-1,6-heptadiene-3,5-dione), chemical structure C_21_H_20_O_6_. Curcumin is a yellow-colored flavonoid with lipophilic characteristics and is water-insoluble.

### 2.2. Phytochemistry, Bioavailability, and Metabolism

The characterization of *C. longa* indicates that its nutritional profile is very complete, that it contains carbohydrates, fiber, some proteins, lipids, vitamin C, pyridoxine, calcium, potassium, magnesium, and phosphorus, and around 235 *C. longa* phytochemicals have been found, the majority of these of a phenolic and terpenoid nature. Curcuminoids comprise the most important phenolic compounds, as around 80% of the total. There are other minor components, such as sesquiterpenes, monoterpenes, diarylheptanoids, triterpenoids, diterpenes, sterols, and alkaloids [20,21]. Curcumin is a curcuminoid chemically identified as a member of the diarylheptanoids. It has a diketonic hydroxycarbon skeleton with a phenolic-conjugated group substituted for by methoxy groups through a 7-atom carbon chain with a 1,3-dicarbonyl function and various unsaturations. In solid phase, it is found in keto form, and in liquid phase as enol (Figure 1). It is a symmetrical structure without stereogenic centers; the skeleton is formed by two phenolic rings connected by a 7-carbon α,β-unsaturated diketone bond with an s hydroxyl group at par around the bond. Curcumin is insoluble in water at a neutral pH and is acidic; it is soluble in organic solvents such as methanol, ethanol, dimethylsulfoxide, and acetone due to its lipophilic nature [22].

Due to its low solubility and chemical characteristics, curcumin appears to be poorly absorbed; it has been reported as having a bioavailability range of between 0.47 and 1%. After the oral administration of 500 mg/kg of curcumin, the maximal concentration (C_max_) in serum was 0.06 µm/mL, and time to reach maximal concentration (T_max_) was 41.7 ± 5.4 min. Elimination half-times (t_1/2,β_) were 28.1 ± 5.6 and 44.5 ± 7.5 min for curcumin (500 mg/kg, per os [p.o.]) and for curcumin (10 mg/kg, intravenously [i.v.]), respectively [23]. At doses of 10 and 12 g, the C_max_ reported was 2.30 ± 0.26 and 1.73 ± 0.19 mg/mL, respectively, while T_max_ and t_1/2_ were 3.29 ± 0.43 and 6.77 ± 0.83 h, respectively Curcumin derivatives such as glucuronide and sulfate were detected in the plasma of human healthy volunteers at 0.25 and 72 h after the consumption of a single oral dose of a curcumin preparation [24]. Another study reported curcumin serum levels of 0.13–1.35 µg/mL at oral doses of 1–2 g/kg. It can be observed that, in spite of its low bioavailability, it is rapidly metabolized and eliminated; the data suggested that curcumin’s low bioavailability can be due to its low water solubility and poor intestinal absorption, an affectation of the intestinal metabolism. The lipophilic nature of the molecule could affect the rate of absorption and, in addition, its rapid systemic metabolism and excretion are important factors [25]. Thus, to improve the bioavailability of curcumin, different strategies have been proven, such as the design of the formulation with nanoparticles, micelles, liposomes, analogues, lipid preparations, etc. [25,26,27,28].

The study conducted in humans demonstrated that a formulation of curcumin with a combination of cellulosic derivates, a hydrophilic carrier, and natural antioxidants (CHC) caused a 45.9-fold-higher absorption of curcuminoids compared with a standardized curcumin mixture; absorption was also significantly improved compared to a curcumin phytosome formulation, by 5.8-fold, and compared to a formulation with volatile oils of turmeric rhizome, by 34.9-fold. The CHC formulation significantly increased curcuminoids in blood in comparison with the other standardized unformulated curcumin preparations [27]. The new γ-cyclodextrin curcumin formulation (CW8) increased plasma concentrations of curcumin 39-fold compared with standardized unformulated curcumin extract, improving curcumin absorption significantly in healthy humans. Ciclodextrin formulations are better absorbed because it forms complexes with lipophilic compounds, improving aqueous solubility and dispersibility [28]. The novel CURCUGEN, a dispersible formulation with 50% curcuminoids—concentrated turmeric extract—demonstrated that the relative bioavailability of total tetrahydrocurcumin was 31 times higher compared to the standard curcumin reference product, curcuminoid 95% standardized extract (C-95). In addition, CURGUGEN had C*_max_* and the area under de curve (AUC0-t) was 16.1 times higher and 39 times higher than that of C-95 [29]. Furthermore, Curcuwin Ultra+ (CU+), a novel formulation designed to protect curcuminoids from intestinal degradation, demonstrated a significant higher total systemic exposure and C_max_ for total curcuminoids compared with 95% turmeric extract (TUR 1800). Besides CU+ was 40% faster absorbed, it possesses superior bioavailability even at lower concentrations (250 and 500 mg) in comparison with higher doses of TUR 1800 (1900 mg) in healthy volunteers under fasting conditions [30]. The water-dispersible turmeric extract containing 60% curcuminoids (TurmXtra 60N), referred to as WDTE60N, using a concentration of 150 mg of curcuminoids, demonstrated a higher absorption and exposure for free curcumin, total curcumin, and total curcuminoids at 10-fold lower dose than standard turmeric extract 95% (STE95) (1500 mg) [31]. Additionally, other substances were utilized to enhance absorption, such as piperine, which increased curcumin bioavailability up to 2000% when administering 2 mg of curcumin plus 20 mg of piperine compared with curcumin alone in rats and humans, with no adverse effects [32]. Piperine improved the bioavailability of curcumin through the reduction of the velocity of its metabolism [33].

Once absorbed, curcumin is biotransformed during phase- and -II reactions by reduction and conjugation reactions in liver. Reduction is performed by NADPH-dependent reductase, alcohol dehydrogenases, and microsomal enzymes. Reduction is carried out in heptadione-chain double bonds to form di- tetra- and octahydrocurcumin. Glucuronidation is performed by the UDP-glucuronosyltransfererases hepatic UGT1A1 and intestinal UGT1A8 and UGT1A10. The isoforms UGT1A9, -1A8, and -2B7 demonstrated high activity for hexahydro-curcuminoids [34]. Sulfathion is produced by the phenol sulfotransferase isoenzymes SULT1A1 and SULT1A3 [26,35]. The major curcumin metabolites identified in vivo are curcumin-glucuronoside, dihydrocurcumin-glucuronoside, tetrahydrocurcumin-glucuronoside, and tetrahydrocurcumin (Figure 2) [36]. It was found that curcumin is converted into some active metabolites by colonic bacteria through the reactions of hydroxylation, demethylation, reduction, acetylation, and demethoxylation. Certain metabolites can recycle through the enterohepatic circulation and are eliminated in urine and feces, but curcumin derivatives are also found in organs such as the brain, liver, spleen, and lung [37].

### 2.3. Curcumin Effects on Muscle Health

#### 2.3.1. Muscle Disorders

Numerous health benefits proven in countless trials have been attributed to curcumin; to date, there has been some evidence that proves these benefits for skeletal muscle (Table 1). For some years, it has been proposed that the use of curcumin may be a viable option for the prevention or therapy of muscle wasting [38]. In particular, in muscle wasting caused by sepsis, curcumin blocks the nuclear translocation of NF-kB subunit p25, as well as p65 DNA binding activity [39]. After traumatic injury, the systematic administration of curcumin induced myogenesis and muscle repair via the inhibition of NF-kB expression [40]. Indeed, curcumin can be considered an inhibitor of the NF-kB pathway, but it can also exert other anti-inflammatory actions within its mechanisms of action, such as inhibiting the activity of the p38 kinase, the induction of the response of heat shock proteins (HSP), and antioxidant activity [41,42].

Several inflammatory conditions in skeletal muscle are related with the upregulation of the AKT and NF-κB and downstream genes such as atrogin-1/MAFbx or MuRF1, as well as TNF-α mediated by p38 MAPK related with the signaling pathways involved in muscle wasting. Jin and Li (2007) previously reported that the administration of curcumin over 4 days (10–60 µg/kg i.p.) attenuated muscle damage induced by lipopolysaccharide (LPS) stimulation in mouse gastrocnemius and extensor digitorum longus (EDL) muscles. Curcumin attenuated LPS-stimulated p38 activation and the upregulation of atrogin-1/MAFbx, inhibiting the loss of muscle mass [43]. However, it must be taken into consideration that this study did not come to express an absolute correlation between the expression of atrogin-1 and MAFbx and the muscle protein content, because an increase was reported of mRNA for atrogin-1, but not for MAFbx.

##### Muscles Affected by Chronic Diseases

Skeletal muscle loss in cancer cachexia appears to be the most relevant clinical issue; unfortunately, it is associated with a poor prognosis. Different models do suggest that muscle waste is the result of reduced protein synthesis and enhanced protein degradation by means of the active ubiquitin–proteosome pathway [53]. Curcumin has proved to interfere in proteosome degradation. In vivo studies found that the curcumin c3 complex at doses of 100 mg/kg body weight (bw) prevents the bw loss in muscle wasting induced in mice by implanting the MAC16 colon tumor. Elevated doses of curcumin c3 (250 mg/kg bw) improved bw gain up to 25% compared to the control. Moreover, there was an increase in muscle fiber size (30–65%) and in the weight of the gastrocnemius muscle (30–58%), and also an inhibition of the expression of muscle-specific ubiquitin ligases as well as the activity of the ubiquitin–proteosome complex; thus, curcumin is effective in preventing and reversing muscle waste, which renders it an optimal aid for the therapy against cachexia [44]. In a randomized, double-blind, placebo-controlled phase-IIa study, 20 patients with the cancer anorexia–cachexia syndrome in locally advanced or advanced head and neck cancer were treated with curcumin (4000 mg/day) or placebo for 8 weeks. The study reported important benefits in improving muscle mass and body composition compared with the placebo. Additionally, hand grip muscle strength and absolute lymphocyte count were favored. It is noteworthy that curcumin was well-tolerated in patients, which is important to bear in mind, due to the gastrointestinal symptoms these individuals tend to exhibit in the progression of and therapy for the disease [45].

Diabetes and other disabilities are related with chronic inflammation mediated by the NF-κB pathway, in which there are an important participation of the proinflammatory cytokines TNF-α and IL-1β. Together with chronic inflammation, elevated oxidative stress is also associated with muscle atrophy, especially in type 1 diabetes mellitus (DM). Streptozotocin-induced type 1 DM in C57BL/6 J mice (200 mg kg^−1^ i.p.) significantly reduced bw, skeletal muscle weight, and CS. Treatment with curcumin (1500 mg kg^−1^ day^−1^) for 2 weeks attenuated the gene expression of the ubiquitin E3 ligase atrogin-1/MAFbx and MuRF-1. Curcumin ameliorated inflammatory markers (cytokines TNF-α and IL-1β) and oxidative stress in type 1 DM. These results indicate that curcumin may be helpful in the management of muscle atrophy in patients with type 1 DM [46]. Other models with dexamethasone, which is a promoter of proteolysis by activating the ubiquitin proteosome system and ROS, point out that the *Curcuma longa*-L. water extract administered in ICR mice at 1 mg/kg/bw day decreased myostatin, MuRF-1, and Atrogin-1 levels along with MDA, and was additionally associated with an increase of antioxidant enzymes that reduced ROS, suggesting that the *Curcuma longa*-L. water extract may be a natural product for preventing skeletal muscle atrophy by regulating muscle atrophy target genes and stimulating an antioxidant response [47].

Chronic obstructive pulmonary disease (COPD) induced by cigarette smoke exposure in combination with LPD in a rodent model demonstrated that curcumin treatment ameliorated damage in airways and improved interstitial fibrosis, myofibril disorganization, inflammation, fiber atrophy, and mitochondrial damage in the skeletal muscle of COPD rats [48]. The mitochondria play a key role in energy production and in the regulation of oxidative stress; in consequence, mitochondrial damage contributes to the development of skeletal muscle dysfunction and the inflammatory response [54]. In the previously mentioned study, a treatment with curcumin significantly improved the mitochondrial enzyme activities of cytochrome c oxidase, succinate dehydrogenase, Na^+^/K^+^-ATPase, and Ca^2+^-ATPase in skeletal muscle. On the other hand, oxidative stress was attenuated by reducing the levels of MDA and enhancing the antioxidant enzymes MnSOD and GPx. In addition, the inflammatory markers IL-6 and TNF-α were reduced. Curcumin promoted the upregulation of the PGC-1α/SIRT3 signaling pathway by increasing mRNA and the protein expression of PGC-1α and SIRT3 of the skeletal muscle of COPD rats [48]. The peroxisome-proliferator-activated receptor gamma coactivator 1 alpha (PGC-1α) is a transcriptional regulator that controls the expressions of the genes involved in mitochondrial biogenesis, energy metabolism, and oxidative stress [55]. The silent-mating-type information regulation 2 homolog 3 (SIRT3) is a downstream gene located in the mitochondrial matrix that participates in mitochondrial fatty acid oxidation and is also active in efficient electron flow in the electron transport chain (ETC) for energy metabolism [56]. It is crucial to mention the role that curcumin might play in the activation of the PGC-1α/SIRT3 signaling pathway; these findings suggest that curcumin could play a pivotal role in the recovery of skeletal muscle in chronic pulmonary disease (COPD).

As with cancer, diabetes, or COPD, chronic kidney disease (CDK) is characterized by a catabolic environment with high levels of oxidative stress and mitochondrial dysfunction, which render these individuals highly susceptible to experiencing muscle atrophy, poor tolerance to exercise, further poor prognosis, low quality of life, and an increase in mortality [57]. Previously, the benefits that apparently accompany curcumin in preventing or reversing muscle atrophy were mentioned, but the complete mechanisms of action remained unclear. Recent findings point out that curcumin treatment (100 mg/kg/bw day) over 12 weeks in a wild-type and muscle-specific glycogen synthase kinase3β (GSK-3β) knockout (KO) CKD mouse model alleviated mitochondrial dysfunction and oxidative damage by inhibiting GSK-3β activity in skeletal muscle [58]. GSK-3β is thought to participate in the regulation of protein metabolism and in mitochondrial function in skeletal muscle [59,60]. In fact, the suppression of GSK-3β is found to be beneficial for PGC-1α signaling and mitochondrial function by increasing the expression of PGC-1α in C2CL2 muscle cells [60]. The genetic and pharmacological inactivation of GSK-3β raises the mitochondrial DNA (mtDNA) copy number, the expression of oxidative phosphorylation (PhoOx) protein levels, and the activity of the enzymatic machinery involved in fatty acid oxidation and the Krebs cycle, enhancing mitochondrial biogenesis during the myogenic process [61]. The researchers observed that curcumin could improve mitochondrial function by optimizing the activity of the electron transport chain and mitochondrial respiration, ATP synthesis, the reduction of the membrane potential, and the attenuation of oxidative mitochondrial stress via the suppression of GSK-3β activity in skeletal muscle [58].

##### Muscles Affected by Toxics

Curcumin could comprise a novel strategy for the treatment of several types of skeletal muscle damage caused by toxic substances such as arsenic. In a model using arsenic trioxide (ATO) to induce muscle damage in ducks, curcumin treatment was able to reduce the oxidative stress manifested by the augmented total antioxidant capacity, SOD, and reduced MDA. Curcumin promoted mitochondrial biogenesis by activating the PGC-1α, NRF1/2, and TFAM pathways. Likewise, curcumin treatment helped to regulate proapoptotic genes (p53, Bax, Caspase-3, and Cytc) and mitophagy (PINK1, Parkin, LC3, and p62) by reducing mRNA protein levels. In turn, curcumin promoted mitochondrial function and integrity [62]. Acute curcumin (1000 mg) together with the oral nutrition supplementation (ONS) of proteins and carbohydrates enhanced the Musculus tibialis anterior microvascular blood volume, and also increased glucose uptake and insulin in the presence of ONS in healthy older adults, which may have an impact on promoting energy metabolism and muscle function [49].

##### Muscles Affected by Immobilization and Sarcopenia

Immobilization during long time periods might induce muscle atrophy, and the proteolytic ubiquitin–proteosome system and the mitochondrial apoptotic pathway play a pivotal role in its development. In the study of Vazeille et al. [63], in which the authors administered curcumin for 8 days to rats subjected to hind limb immobilization, the results indicated that curcumin did not reduce muscle atrophy at 8 days of immobilization, but instead improved muscle recovery and the CSA of immobilized muscles after 10 days. The authors observed that curcumin hampers proteasome chymotrypsin-like activity and the trend toward increased caspase-9-associated apoptosome activity at 8 days of immobilization. In addition, curcumin improved muscle recovery during reloading due to the reduction of the protein levels associated with immobilization (Smac/DIABLO) and the elevation of X-linked inhibitory apoptotic proteins after 10 days, enhancing muscle regeneration during the first steps. Recently, the use of curcumin (1 mg/kg/bw i.p.) and resveratrol (20 mg/kg/bw i.p.) elicited important changes in the number of muscle satellite cells, as well as progenitor muscle cell numbers, activating the quiescent cells in limb muscles of mice (C57BL/6J, 10 weeks) subjected to reloading for 7 days subsequent to a 7-day period of hind limb immobilization [50]. Curcumin alone promoted the growth of CSA and the recovery of muscle fibers, and increased the activity of sirtuin 1. Treatment with both curcumin and resveratrol promoted the numbers of subtypes of satellite cells in the unloaded limb muscle, but not in the reloaded muscle. Therefore, the benefits may not be equal in the reloaded phase; however, the target phytochemicals could have potential clinical advantages during muscle regeneration. Certain other pathways involved in immobilization-induced atrophy have improved when curcumin is administered. For instance, the reduction of proteolytic and signaling markers (NF-kB p50) was reported; conversely, sirtuin-1 and hybrid fiber size attenuated in the gastrocnemius of mice subjected to reloading following 7 days of immobilization. It was hypothesized that curcumin would attenuate muscle proteolysis by the activation of histone deacetylase sirtuin-1, decreasing the atrophy signaling pathways [64].

Another study tested the combination of curcumin (1% diet) and fish oil (5% diet) to determine its effect on anabolic signals and protector stress proteins in C57BL/6 mice subjected to hind limb unloading. The results indicated that the intake of fish oil and curcumin for 10 days prior to hind limb unloading improved the CSA and enhanced the anabolic signaling of Akt phosphorylation, p70S6K phosphorylation, and the abundance of HSP70 while simultaneously reducing the NADPH oxidase-2-complex (Nox2), an indicator of oxidative stress, suggesting that such a mixture could aid in preventing muscle atrophy [65]. Recent available data highlight that the suppression of Nox2 would attenuate the disruption of HPS70, sarcolemnal oxide nitric synthase (nNOS), and Nrf2, consequently mitigating unloading-induced muscle atrophy [51].

Curcumin is also being associated with the prevention of age-related sarcopenia. Gorza et al. [52] evaluated the effect of curcumin treatment at 120 µg/kg in a volume of 100 µL in 18-month-old C57BL6J and C57BL10ScSn male mice. In this study, the researchers observed that curcumin was able to significantly increase survival in both strains, preventing sarcopenia and pre-sarcopenia, preserving type-1 myofibers, and increasing type-2A ones, positively influencing the satellite cells by preserving adult levels of myofiber maturation in old, regenerating soleus muscle.

#### 2.3.2. Physical Exercise and Skeletal Muscle 

Little evidence has been produced by studies conducted under exercise conditions to prove the effects of curcumin on protecting muscle from exercise-induced damage. Recently, it was reported that curcumin exerts a positive impact on performance during exercise and may enhance recovery. A new formulation of curcumin, Next-Generation Ultrasol Curcumin (NGUC), which is more bioavailable, was evaluated in a rodent exhaustive treadmill exercise protocol. Animals that received NGUC at doses of 100 and 200 mg/kg/bw improved endurance capacity and hand grip strength; at the same time, indicators of fatigue (lactic acid, LDH), muscle damage (creatine kinase) (CK), oxidative stress (MDA), and inflammation were attenuated. In particular, IL-1β, IL-6, and TNF-α proteins in muscle were effectively reduced in the exercise group at doses of 200 mg/kg of NGUC. The study indicated that NGUC reduced mTOR phosphorylation similarly to that of the PGC-1α protein level and diminished MAFbx and MuRF1 protein levels. In conclusion, curcumin promoted muscle recovery and exercise performance, attenuating muscle damage and activating anti-inflammatory, antioxidant, and muscle-mass regulatory signaling in a dose-dependent manner [66]. In another eccentric exercise model, curcumin was administered over 20 days via oral gavage at doses of 200 mg/kg bw dissolved in corn oil. On day 21, the rats were subjected to a treadmill run and sacrificed; the analysis of blood and muscle tissue demonstrated that curcumin attenuated the muscle damage induced by the eccentric exercise. However, a difference was not observed in the antioxidant response because there were no significant differences in SOD and glutathione and MDA levels. Thus, it can be concluded that curcumin protected against muscle damage but did not necessarily exert an impact on oxidative stress and the antioxidant response [67].

In an exercise protocol, 6-week-old treadmill-running rats were exercised at 25 m/min for 45 min following a physical adaptation period. Animals were administered with 20 mg of curcuminoids daily during the 6-week exercise protocol. The results revealed significant improvement in run-to-exhaustion time and a reduction of MDA levels, NF-κB, and HSP70 in animals treated with curcuminoids. Furthermore, the protein levels of Sirt1, PGC-1α, Nrf2, and GLUT4 increased significantly, and the antioxidant–enzyme activities of GPx and SOD were higher, similar to those of the glutathione content compared with non-treated animals. In conclusion, curcumin protected skeletal muscle from the damage induced by exhaustive exercise, inhibiting NF-κB and promoting Nrf2 signaling pathways [68]. In another model of heart failure with a low ejection fraction (HFrEF)—a mouse model with ligation-induced coronary artery characterized by intolerance to exercise—the application of subcutaneous osmotic minipump curcumin (50 mg·kg^−1^·day^−1^ for 8 weeks) was evaluated. It was found that animals treated with curcumin significantly improved in maximal speed, running distance to exhaustion, and limb grip force. It could also be observed that reduced force and rapid fatigue in soleus and extensor digitorium longus muscles were reduced. Likewise, curcumin enhanced Nrf2, SOD, hemoxygenase-1 (HO-1), MyoD, and myogenin; these results indicated a positive impact on reducing oxidative stress, promoting exercise performance, and better tolerance to exercise in HFrEF mice [69].

The underlying mechanism of curcumin regarding muscle health is summarized in Figure 3.

## 3. Sulforaphane

### 3.1. Botanical Description

Brassicaceae or Cruciferae is a family that includes various cruciferous vegetables, such as broccoli, cauliflower, cabbage, kohlrabi, mustard, and brussels sprouts. Broccoli (*Brassica oleracea* L. *var. italica* Plenk) is 45.7–76.2 cm in height, with a 30–60-cm spread. The stem arises from the roots and is surrounded by leaves; mature broccoli flowers are small with four yellow petals [70]. Broccoli contains fruits in siliqua with rounded pink seeds. Broccoli has green leaves and a green flowerhead (the edible inflorescence) that bears green, purple, yellow, or white flowers. Broccoli leaves have petioles with elongated limbs with grey-green leaves and wavy deep lobes. Broccoli may be grown as an annual, as a biennial (flowers in the second year), or as a perennial crop, depending on the type of broccoli and the region. It is a harvest-time seasonal bloomer and needs full sun and a medium amount of water to grow. It is a cool-weather vegetable that is grown for a harvest of large, tight terminal heads of green flower buds at the ends of thick, edible stems [71]. *Brassica oleracea* is thought to be the phylogenetic parent of broccoli; it is a species native to Atlantic coastal Europe, and it occurs along the coasts of the United Kingdom, Germany, France, and Spain. Brassica species have been cultivated for at least 2000 years [72].

Broccoli is highlighted because of its content of the bioactive compound sulforaphane (SFN). Several studies have been conducted with regard to the consumption of cruciferous vegetables and their multiple benefits to health, supporting that the bioactive compounds that they possess are responsible for their previously mentioned biological activities. The majority of studies support the potential anticancer, antidiabetic, and cardioprotector effects, and its help in losing body weight (bw). Since 1992, Zhang et al. [73] reported that SFN and its sulfide and sulfone analogues induced the activities of the phase II detoxification enzymes NQO1 and GST in several mouse tissues, in that it is known that SFN is the most biologically active component deriving from broccoli with anticarcinogenic action.

### 3.2. Phytochemistry and Bootability and Metabolism

Glucosinolates (GSL) are the major biologically active compounds within *Brassicaceae* species; they are the precursors of isothiocyanates (ITC), which are produced by myrosinase during slicing, harvesting, and chewing by enzymatic degradation [74]. GSL are thiglucosides that share a β-D-thioglucose, a sulfonated oxime group, and a side chain derived from some essential amino acids (such as phenylalanine, tryptophan, and methionine) as their basic structure [75] (Figure 4). In the human digestive tract, GSL can be transformed by the intestinal microflora myrosinase isoform. SFN is the most studied among the GSL, which currently entertains the major evidence of beneficial effects. The SFN chemical structure comprises 4-methylsulfinybutyl isothiocyanate; young broccoli contains approximately 1153 mg/100 mg of SFN and the mature vegetable contains 44–171 mg/100 mg dry weight (dw). The edible portion of mature broccoli contains 507–684 µg/g SFN dw [76]. SFN is lipophilic in nature, of a low molecular weight, and is a thermo-sensitive molecule [77].

The bioavailability of SFN depends on certain factors, such as mode of preparation. For the cooking of broccoli, for example, it has been reported that on quickly steaming the broccoli sprouts following a myrosinase treatment, the vegetable contains 11 and 5 times higher amounts of SFN than freeze-dried and untreated steamed broccoli. After the oral administration of 2.5 mg/g bw of the broccoli sprout preparations (quickly steaming and unsteaming), SFN was rapidly absorbed and distributed throughout the tissues. The SFN-rich preparation presented the highest peak of a plasma SF concentration of 337 ng/mL, which is 6.0 times and 2.6 times higher compared than that of the other two preparations [78]. The oral administration of SFN 2.8 µmol/kg or 0.5 mg/kg presents 80% bioavailability, whereas doses of 28 µmol/kg or 5 mg/kg demonstrated only around 20% bioavailability [79]. In another study, a peak concentration was found of SFN metabolites of 1.91 ± 0.24 µM after 1 h with an oral dose of 200 µmol. The same oral dose of 200 µmol reported a peak concentration of SFN metabolites of 0.7 ± 0.2 µM after 3 h [80]. The consumption of 200 g of crushed broccoli indicated that higher bioavailability was found (37%) in men who ate raw broccoli vs. men who consumed cooked broccoli (bioavailability, 3.4%). It was found that the time of absorption was delayed in cooked broccoli (peak plasma time = 6 h) compared with raw broccoli (1.6 h); nevertheless, their excretion half-lives were similar, that is, 2.5 and 2.4 for raw and cooked broccoli, respectively [81].

After the conversion of glucosinolates by myrosinase, SFN is metabolized within the body by conjugation in the presence of glutathione. The reaction produces N-acetylcysteine derivatives in the form of mercapturic acids, termed dithiocarbamates (DTC), or conjugates of SFN through the mercapturic acid pathway. First, SFN is conjugated with glutathione in a glutathione transferase GST catalyzed reaction. Afterward, sulforaphane-N-acetylcysteine (SFR-NAC) is systematized via cleavage reactions catalyzed by *γ*-glutamyl transpeptidase, cysteinyl glycinase, and *N*-acetyltransferase [75]. Following the oral consumption of 200 µmol of SFN from broccoli sprouts, 70–90% of DTC metabolites were identified in urine [82]. According to the pharmacokinetic assessment of the relation of dithiocarbamates (DTC)/isothiocyanates (ITC) in human volunteers who received 200 μmol of broccoli sprout isothiocyanates (SFN at a major proportion and less iberin and erucin), it was found that SFN was rapidly absorbed (concentration 0.943–2.27 μmol/l) in plasma, serum, and erythrocytes 1 h after intake, with a reported half-life of 1.77 ± 0.13 h and an excretion time at 8 h (58.3 ± 2.8% of the dose). Renal clearance was 369 ± 53 mL/min [83].

### 3.3. Sulforaphane Effects on Muscle Health

#### 3.3.1. Sulforaphane and Biological Activity

SFN is considered the most powerful nutraceutical contained in cruciferous vegetables, due to its multiple biological activities on and benefits to human health described in a growing body of evidence. The anticarcinogenic activity of SFN is a very efficient activator of the E2 factor-related factor (Nrf2) and the signaling pathway involved in the antioxidant and cytoprotector response in response to stress stimuli [84,85,86]. The activation of Nrf2 is closely involved in the repression of NF-ΚB inflammatory signaling and downstream proinflammatory cytokines and their mediators [87]. Nrf2 is a member of the cap ‘n’ collar (CNC) family of basic region–leucine zipper (bZIP) transcription factors. In humans, Nrf2 is a 66-kDa and 606-amino-acid protein divided into Nrf2-ECH homology regions and seven Neh domains; it is coded by the *NFE2L2* gene. These factors regulate more than 250 genes implicated in cell defense and in the redox response [88].

Under homeostatic circumstances, Nrf2 remains attached in the cytosol to the suppressor regulator Kelch-like erythroid cell-derived protein with CNC homology-associated protein 1 (Keap1) through the ETGE and DLG motifs in the Neh2 domain, forming a dimer with Keap1 in Kelch domains, leading to the ubiquitination of seven lysines and the consequent degradation in the 26S complex [89]. Keap1 is a redox-sensitive regulator and dissociates from Nrf2 with electrophilics such as ROS/NOS. Thus, Nrf2 is released and translocated into the nucleus to bind the specific DNA sequence of the antioxidant response element (ARE) within small musculo-aponeurotic fibrosarcoma proteins (sMaf) [90]. Bonding with the ARE sequence promotes the expression of a wide range of genes involved in cytoprotection and the antioxidant response, such as the phase II detoxifying enzymes NAD(P)H quinone oxidoreductase 1 (NQO1) and glutathione S-transferase (GST), heme oxygenase 1 (HO-1), and enzymes involved in the synthesis and metabolism of glutathione. In agreement with the latter, Nrf2 modulation has become a target for elucidating the mechanism of several aliments related with redox balance [91]. It has been found that different synthetic or natural compounds are potential Nrf2 activators; SFN efficiently activates the Nrf2/Keap1/ARE signaling pathway. It has been discovered that more than 500 genes are activated by SFN through the Nrf2/ARE signaling pathway; for that reason, SFN is considered a potent nutrigenomic compound. SFN can induce Nrf2 translocation and nuclear accumulation and can phosphorylate Nrf2 through the activation of MAPK, protein kinase B (PKB/Akt), and protein kinase C (PKC) [92,93].

From the nutrigenomic perspective, certain molecules, including some phytochemicals, are effective Nrf2 activators or inducers according to their CD value, which is an indicator of the amount of specific compound required to double NQO1 activity in murine hepatoma cells (hepg2). NQO1 is one of the most important protector enzymes and it has been employed to evaluate the phytochemicals involved in chemopreventive and anticancer activity [94]. The nutrigenomic effect of certain bioactive compounds can be determined by measuring the CD value; SFN exhibited the highest potential, taking into account that the lesser the amount of the concentration required to activate Nrf2, the more efficient an activator the compound is considered to be. SFN requires 0.2 μM and curcumin, 2.7 μM; this may help towards understanding why both SFN and curcumin have achieved very promising results in numerous trials. Among other effective phytochemicals reported, we find andrographolides, silymarin, quercetin, beta-carotene, genistein, lutein, resveratrol, and zeaxanthin [95]. Substantial evidence from pre-clinical and clinical reports supports that SFN deriving from cruciferous vegetables such as broccoli possesses enormous chemoprotective potential and more benefits to health upon its use [96].

In addition to the activation of the antioxidant response, SFN is capable of modulating the inflammatory response by inhibiting the binding of NF-κB to DNA. Furthermore, it constrains the activation of I-κB and the translocation of NF-κB, thereby reducing inflammation. Suppression of the activation of the NF-κB signaling pathway blocks the release of inflammatory mediators such as TNF-α IL-1β, IL-6, nitric oxide (NO), and prostaglandin E2 (PGE2). The attenuated activity can be observed in inflammatory enzymes such as cyclooxygenase-2 (COX-2) and inducible NO synthase (iNOS).

#### 3.3.2. Muscle Disorders

The cumulative studies conducted of SFN on skeletal muscle are presented in Table 2. Since SFN is described as one of the major bioactivators of the Nrf2/Keap1/ARE signaling pathway, some studies report that it protects skeletal muscle from damage in different models; however, evidence of this remains scarce. Duchenne muscular dystrophy is a muscle disorder associated with elevated oxidative stress and inflammation. The investigation was conducted by Sun et al., who utilized a rodent Duchenne muscular dystrophy model. The animals were treated with SFN by gavage (2 mg·kg body wt^−1^·day^−1^ for 8 weeks. SFN treatment augmented muscle mass, strength, and running capacity related with the enhancement of subsarcolemmal integrity, central nucleation, and myofibrillar variability. Likewise, the GSH/GSSG ratio was favorable; the muscle markers of damage and oxidative stress, LDH, CPK, and MDA, were significantly reduced [97]. Afterward, in a very similar model, it was evidenced that SFN activated Nrf2 and targeted the expression of the HO-1 enzyme; in parallel, the central inflammatory-signaling command by NF-κB was inhibited with the reported reduction of NF-κB (p65), and the phosphorylated IκB kinase-α augmented inhibitor κB-α expression, proinflammatory cytokines TNF-α, IL-1β, and IL-6, and the inflammatory cytokine CD45, as long as the infiltration of immune cells into mdx mouse skeletal muscle lasted. In summary, SFN activates the antioxidant response and blunts inflammatory signaling; therefore, it may be a potential tool with clinical benefits in the therapy of individuals with muscular dystrophy [98].

Beyond the scope of whether SFN would modulate antioxidant and inflammatory responses, new research conducted by Zhan et al. [99] reported that SFN can control the growth of porcine satellite cells (PSC) and that it epigenetically increased the expression of SMAD7, a family member of regulators involved in myogenesis and muscle regeneration that inhibits transforming growth factor beta (TGF-β) signaling. SFN at 5, 10, and 15 µM boosted PSC proliferation by modifying the mRNA expression of myogenic regulatory factors. Moreover, SFN repressed histone deacetylase (HDAC) activity and disturbed mRNA levels of HDAC family members, favoring an abundance of histone H3 and H4 in PSC. SFN improved the level of acetylation of histone H4 in the SMAD7 promoter; at the same time, SFN reduced the expression of microRNA. These results indicate that SFN may be a powerful stimulator of skeletal muscle growth. Previous work demonstrated that SFN (5 μM, 10 μM, and 15 μM) avoids oxidative stress and apoptosis in PS; further, SFN acts as an HDAC and DNA methyltransferase (DNMT) inhibitor. The most relevant issue was related to the inhibition of myostatin expression and the markedly lesser expression of the negative feedback inhibitors of myostatin signaling [104]. Myostatin is a member of the TGF-β family and is considered a potent inhibitor of skeletal muscle growth; it can hamper cell activation and the cell renewal of satellite cells [105]. Hence, SFN can be a nutraceutical that can inhibit the myostatin signaling pathway, promoting muscle anabolism.

#### 3.3.3. Physical Exercise and Skeletal Muscle

Cumulative evidence supports that the practice of regular physical exercise provides great benefits to health and that, in general, it prevents the development of chronic diseases, improves immune function, and prevents obesity. Nevertheless, extended exercise sessions along with inadequate recovery periods may cause muscle damage. Throughout exercise, the increased oxygen consumption induces an elevated production of ROS within the myofibers, which is counteracted by antioxidant defenses; nonetheless, overtraining may surpass the ability to constrain exercise-induced oxidative stress [106]. The prolonged periods of fatigue induced by intense training or competitive seasons, along with inappropriate recovery post-exercise/competition episodes, contributes to the development of overtraining syndrome (OTS). Consequently, excessive ROS and OTS are closely related with muscle fatigue and low physical performance [107]. Muscle fatigue can be defined as a decline in maximal force production in response to contractile activity [108]. Detrimental skeletal muscle force could also be explained by the predominance of the oxidized state of muscle fibers continuously exposed to elevated ROS, especially H_2_O_2_ [109]. In contrast, unfatigued muscle fibers are maintained in a reduced state. Furthermore, inflammation is also a result of the inability to restrain the sustaining of redox imbalance. In exhaustive exercise, inflammatory cells such as macrophages and neutrophils infiltrate into muscle fibers. The released cytokines, chemokines, and damage-associated molecular patterns (DAMP) are increased in damaged tissue, which promotes the migration of leukocytes [110].

SFN is considered the most powerful nutraceutical contained in cruciferous vegetables due to the multiple biological activities described in this efficient activator of the Nrf2 pathway. SFN may prevent damage to skeletal muscle during very hard physical workouts. Previously, SFN was reported at doses of 25 mg/kg/bw i.p. in male Wistar rats that performed a single bout of exercise until exhaustion on a rodent treadmill (+7% slope and 24% slope). The 3-day SFN treatment increased the expression and activity of glutathione S-transferase (GST), glutathione reductase (GR), and NQO1; the expression of Nrf2 augmented significantly, as did total antioxidant capacity. On the other hand, a decrease was found in LDH and CK activity. These results suggest that the SFN pre-treatment may exert a protective effect against muscle damage induced by exhaustive training, the mechanism of action appearing to be that SFN triggers the antioxidant pathway [100]. Afterward, wild-type mice (Nrf2+/+) and Nrf2-null mice (Nrf2−/−) C57BL/6J were subjected to a progressive continuous all-out test. The administration of SFN (25 mg/kg/bw i.p.) significantly improved the run distance, which was directly related with the upregulation of the Nrf2 target genes *NQO1*, *OH-1*, *CAT*, and *ƴ-GCS* in the gastrocnemius muscle and soleus of SFN-treated Nrf2+/+ mice. Of note, CK and LDH as muscle damage markers significantly decreased, and the lactate content remained low in the same group. Indicators of oxidative damage, such as TBARS, were reduced as well. Additionally, a higher number of mtDNA was also reported, but intriguingly, there were no significant differences in PGC-1α and Sirt1, though AMPKα increased. The authors suggested that SFN exerts a protective effect on active skeletal muscles, preventing fatigue, attenuating oxidative damage, and improving aerobic capacity through the upregulation of the antioxidant response during exhaustive exercise [101].

The studies reporting the use of SFN in terms of exercise and muscle recovery on humans are very limited. Notwithstanding this, lately the manner in which SFN could impact muscle soreness was explored in young men who performed six sets of five eccentric exercises with the nondominant arm in elbow flexion with 70% maximal voluntary contraction by assessing the evaluation of delayed-onset muscle soreness (DMSO) and range of motion (ROM). After 2 weeks of SFN supplementation (30 mg/day), a marked reduction was observed in muscle soreness associated with a reduction of MDA serum levels, whereas the mRNA expression of NQO1 increased following 2 days of exercise, suggesting that SFN could aid in muscle recovery [102]. The study of Ruhee et al. [103] evaluated the oral consumption of SFN (50 mg/kg bw) 2 h prior to a running treadmill test. Acute exhaustive exercise elevated the damage markers alanine aminotransferase (ALT), aspartate aminotransferase (AST), and LDH. Moreover, a significant increase was observed in the mRNA expression of the proinflammatory cytokines IL-6, IL-1β, and TNF-α in the livers of the exercise group. However, SFN treatment remarkably reduced the biomarkers of tissue damage and cell death. Along with the latter, SFN upregulated Nrf2 signaling by increasing the mRNA expression of Nrf2, HO-1, and the antioxidant enzymes SOD1, CAT, and GPx1 in the liver of SFN-treated animals.

The underlying mechanisms of SFN with respect to muscle health are summarized in Figure 5.

## 4. Conclusions and Perspectives

The progressive deterioration of skeletal muscle mass can lead to the development of frailty and immobility in elderly individuals or in those suffering from chronic diseases such as cancer, diabetes, chronic kidney disease, AIDS, or severe trauma. The decrease in locomotion of the individual engenders dependency and a poor quality of life, in addition to generating a great burden for the economic budgets of health systems worldwide. Therefore, it is necessary to explore different medical alternatives to ascertain whether they are of aid in preventing deterioration or serve in the treatment of diseases that induce muscle damage or atrophy, promoting recovery and the maintenance of same. Since ancient times, the use of plants or foods has formed part of traditional medicine to improve human health. In this regard, it is known that there are endless species with specific therapeutic uses for certain conditions, which have been formally studied in order to find those responsible for their therapeutic effects. Based on this, it is known that there are active compounds in plants and foods with biological activity, denominated phytochemicals. Curcumin from *Curcuma longa* Linn and SFN from *Brassicaceae* species have been studied in several trials to prove their potential as nutraceuticals that are able to provide multiple benefits for human health.

In different models of muscle damage, including pathological or exhaustive exercise, curcumin and SFN have proven to be effective in preventing or reducing injuries to skeletal muscle mass by their investment in the promotion of the signaling pathways involved in cytoprotection and optimal antioxidant response. Both bioactive compounds efficiently blunt inflammation and help to recover skeletal muscle. In particular, curcumin has shown to prevent muscle atrophy by the suppression of protein synthesis, mostly by the downregulation of ubiquitin ligases and by the promotional, myogenetic, and mitochondrial qualities demonstrated in its in vitro and in vivo studies. A limitation of the use of phytochemicals lies in that the targeted studies employ supraphysiological amounts that are hardly achieved under normal circumstances with the foods containing them. The issue of bioavailability must also be considered, as in the case of curcumin, but not with SFN, contained in broccoli, in which other factors are involved, such as preparing, cooking, etc. Finally, more studies should be conducted in humans because, although research on cell lines or in animal models is quite useful for understanding the molecular mechanisms by which active compounds act, it is necessary to understand their effects on human biological systems.

## Figures and Tables

**Figure 1 plants-11-02517-f001:**
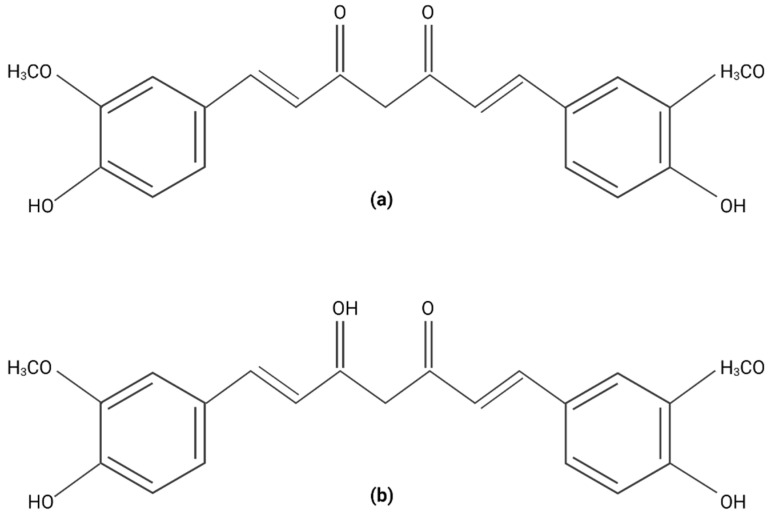
Curcumin chemical structure. (**a**) Ceto form. (**b**) Enol form. Created with BioRender.com.

**Figure 2 plants-11-02517-f002:**
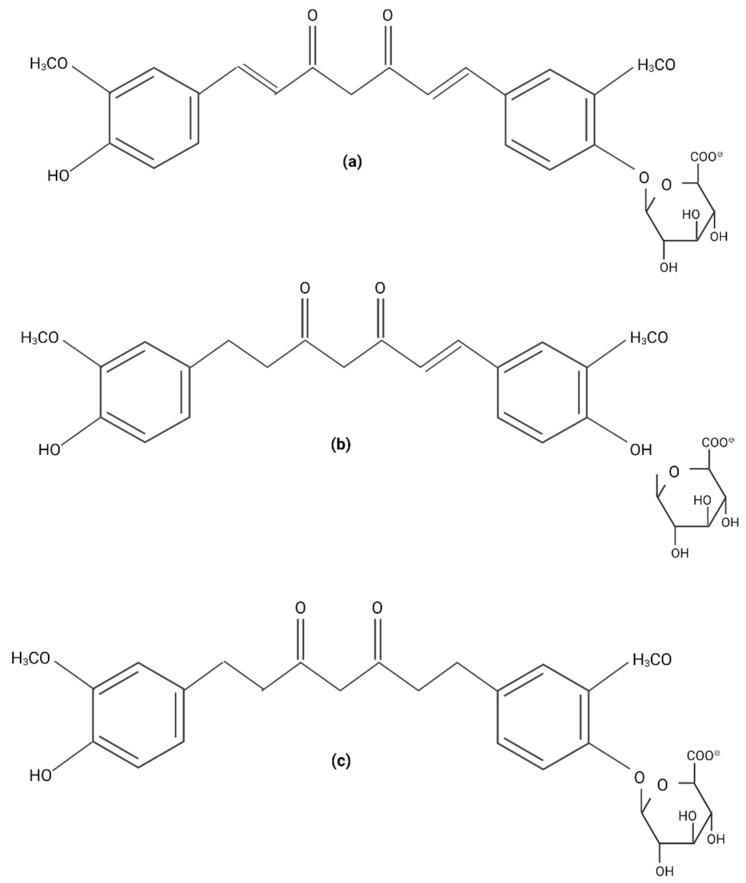
**Curcumin metabolites**. (**a**) Curcumin-glucurunoside. (**b**) Dihydrocurcumin-glucurunoside. (**c**) Tetrahydrocurcumin-glucurunoside.

**Figure 3 plants-11-02517-f003:**
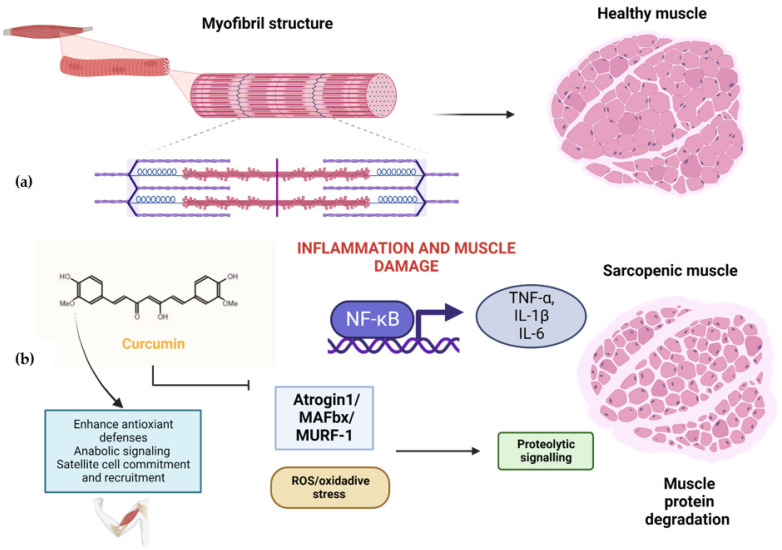
(**a**) Myofibril structure in healthy muscles. (**b**) Effects of curcumin on human skeletal muscle. Curcumin inhibits inflammation and muscle damage by hampering NF-kB and the proinflammatory interleukins TNF-α, IL-1β, and IL-6, as well as the proteasome complex system for protein muscle degradation integrated by ubiquitin ligases atrogin 1, muscle atrophy F-box (MAFbx), and muscle RING finger-1 (MUFR-1). Curcumin reduces oxidative-stress-enhancing antioxidant cell defenses and promotes anabolic signaling, myofibril integrity, mitochondrial function, and satellite cell commitment and recruitment for muscle repair. Created with BioRender.com.

**Figure 4 plants-11-02517-f004:**
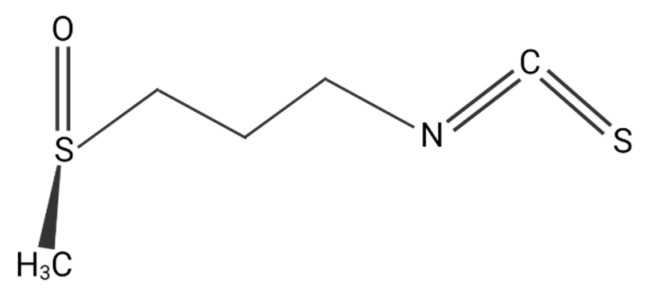
Sulforaphane, basic chemical structure. Created with BioRender.com.

**Figure 5 plants-11-02517-f005:**
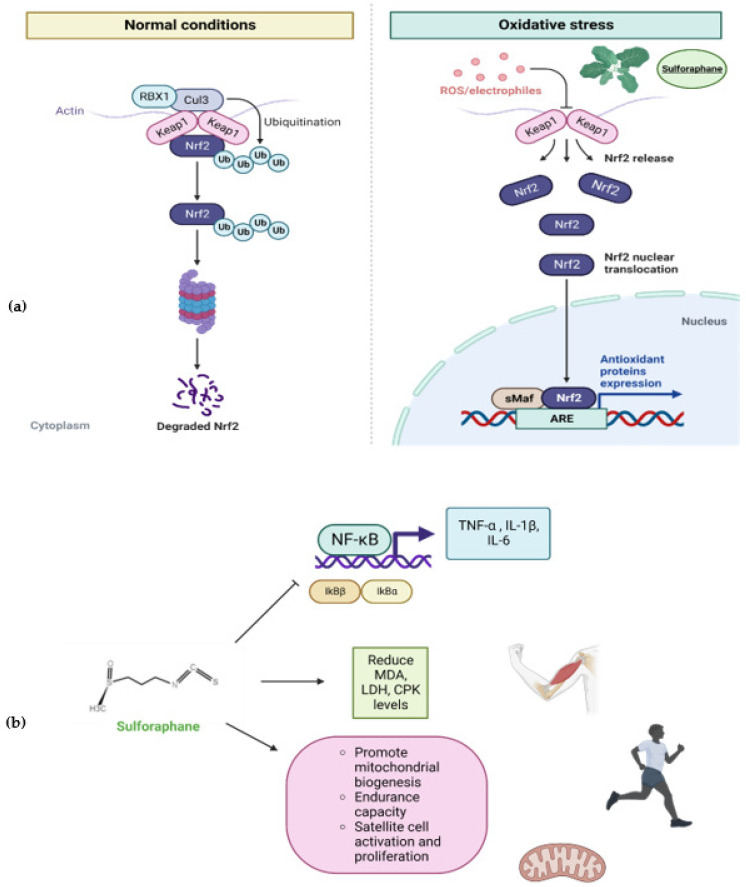
(**a**) Sulforaphane effects on Nrf2 signaling pathway. Sulforaphane promotes dissociation of Nrf2 from its negative regulator Keap1 leading to nuclear translocation and binding to the specific DNA sequence antioxidant response element (ARE) within small musculo-aponeurotic fibrosarcoma proteins (sMaf) inducing the antioxidant protein expression. (**b**) Effects of sulforaphane on skeletal muscle. Sulforaphane inhibits NF-κB inflammatory signaling and reduces muscle oxidative damage biomarkers such malonaldehyde (MDA), lactate dehydrogenase (LDH), and creatin phospho-kinase (CPK). Sulforaphane is capable of promoting mitochondrial biogenesis, improving aerobic endurance capacity, and enhancing satellite cell activation and proliferation. Created with BioRender.com.

**Table 1 plants-11-02517-t001:** Studies evaluating the effect of curcumin in several models of muscle disorders.

Model	Supplementation	Curcumin Effects	Reference
In vivo: skeletal muscle wasting model in mice induced by LPS	Daily i.p. injection of curcumin (10–60 µg/kg) for 4 days	Inhibition of LPS-stimulated p38 activation and upregulation of atrogin-1/MAFbx in gastrocnemius and EDL muscles blocking loss of skeletal muscle mass	[43]
In vitro: human skeletal muscle cells In vivo: implant of MAC16 colon tumor in mice to induce muscle wasting	Curcumin c3 complex 2–5 μg/mL in muscle cells 100–250 mg/kg bw in mice	Inhibition of tyrosine release and chymotrypsin-like 20S proteasome activity muscle cells Prevention of weight loss at low doses (100 mg/kg bw) Promotion of body weight at high doses (250 mg/kg bw) Increase of muscle fiber size Inhibition of proteasome complex activity Reduced expression of ubiquitin ligases MAFbx and MURF-1	[44]
Individuals with CAS, randomized double-blind study	Oral curcumin dose 4000 mg/kg bw daily/8-weeks	Improved muscle mass, body composition, and handgrip strength Changes in absolute lymphocyte count	[45]
In vivo: skeletal muscle atrophy model in C57BL/6 J mice SZT-induced type 1 DM	Diet with or without curcumin 1500 mg/kg bw/day/2-weeks	Decrease in ubiquitination protein Reduced gene expression of muscle ubiquitin E3 ligase MAFbx and MURF-1 Inhibition of activation and concentration of NF-κB, IL-1β and TNF-α	[46]
In vivo: DEX muscle atrophy model in ICR mice	CLW 1 g/kg bw/day by gavage 1 week before DEX injection	CLW helped to suppress the decrease in handgrip strength Inhibition of decrease in muscle mass Inhibition of mRNA expression of myostatin, MURF-1 and atrogin-1 Improved antioxidant activity SOD, CAT, GPx and reduction of MDA levels	[47]
In vivo: COPD model in Sprague Dawley rats	Oral curcumin administration	Improved muscle fiber atrophy, myofibril disorganization, mitochondrial structure, and interstitial fibrosis Enhancement of mitochondrial enzyme activity, antioxidant enzymes MnSOD, GPx, and CAT Attenuation of MDA, IL-6, and TNF-α Increased mRNA expression of PGC-1α and SIRT-3	[48]
Healthy older adults: evaluation of macro and microvascular function, endothelial function, insulin and glucose metabolic response	Acute curcumin 1000 mg with and ONS vs. placebo	Improvement of MBV in m. tibialis anterior without potentiating m. vastus lateralis MBV, glucose uptake and endothelial or macrovascular function	[49]
In vivo: reloading and immobilization model in female C57BL/6J mice	Immobilization period: curcumin 1 mg/kg bw/24 h i.p./1 to 7 day Recovery period: curcumin 1 mg/kg bw/24 h i.p./day 8 to 14 day	Inhibition of proteolytic and signaling markers NF-κB, decrease of SIRT-1 Increase of fiber size of reloaded muscles Attenuated proteolysis via activation of deacetylation of SIRT-1 and decrease of atrophy signaling	[50]
In vivo: HU model in C57BLC mice	Administration of 5% fish oil and 1% curcumin in diet 10 days prior to HU	Improvement of muscle CSA and abundance of HSP70 Enhancement of anabolic signaling phosphorylation of Akt and p60S6K Reduced Nox2	[51]
In vivo: aging presarcopenia/sarcopenia in C57BL6J and C57BL10ScSn male mice	120 μg/kg of curcumin formulation in a volume of 100 μL s.c./6 months	Increase of survival in both strains without signs of liver toxicity Prevention of sarcopenia in soleus and presarcopenia in EDL Preservation of type 1 myofiber size, increase type 2A one in soleus muscle Improvement in satellite cell commitment and recruitment	[52]

i.p.: intraperitoneal; NF-κB: nuclear factor kappa B; LPS: lipopolysaccharide; EDL: extensor digitorium longus; MAFbx: atrogin-1/muscle atrophy F-box; MUFR-1: muscle RING finger-1; CAS: cancer anorexia–cachexia syndrome; SZT: streptozotocyne; DM: diabetes mellitus; DEX: dexamethasone; CLW: *Curcuma longa* water extract; SOD: superoxide dismutase, CAT: catalase; GPx: glutathione peroxidase; MDA: malonaldehyde; COPD: chronic obstructive pulmonary disease; IL-6: interleukin-6; PGC-1α: peroxisome proliferator-activated receptor-gamma coactivator; SIRT-3: sirtuin 3; ATO: arsenic trioxide; ONS: oral nutritional supplement; MBV: microvascular blood volume; WT: wild type; KO: knock out; SIRT-1: sirtuin 1; HU: hindlimb unloading; CSA: cross sectional area; Nox2: NADPH oxidase 2; s.c.: subcutaneous.

**Table 2 plants-11-02517-t002:** Studies evaluating the effect of sulforaphane in several models of muscle disorders and exercise.

Model	Supplementation	Sulforaphane Effects	Reference
In vivo: Duchenne muscular dystrophy in mdx mice	SFN 2 mg/kg bw/day by gavage/8 weeks	Increased expression and activity of NQO1 and HO-1 in dependent manner of Nrf2 Increase in skeletal muscle mass, muscle force, running distance and GSH-to-GSSG ratio Decreased activities of CK and LDH Reduction in MDA levels, myocardial and gastrocnemius hypertrophy Improved fiber size ability, inflammation and sarcolemma integrity	[97]
In vivo: muscle dystrophy in mdx mice	SFN 2 mg/kg bw/day by gavage/4 weeks	Increased expression and activity of HO-1 IN dependent manner of Nrf2 Decreased inflammation, cell infiltrate, proinflammatory cytokine CD45 and inflammatory cytokines TNF-α and IL-1β Reduced expression of NF-κB(p65) and phosphorylated IκB kinase-α Increased IκB-α expression in Nrf2-dependent manner	[98]
In vitro: porcine satellite cells	SFN at 5, 10, and 15 µM	Inhibition of HDAC activity and disturbed mRNA levels of HDAC family members Elevation of acetylated histone H3 and H4 Upregulation of protein and mRNA levels of SMAD7 Increased the acetylation level of histone H4 in the SMAD7 promoter Increased PSC proliferation	[99]
In vivo: male Wistar rats Acute exhaustive exercise	SFN pre-training 25 mg/kg bw i.p.	Enhanced Nrf2 expression and the downstream target genes NQO1, GST, and GR in vastus lateralis muscle Promoted antioxidant enzyme activity Increased TAC Reduced LDH and CK activities	[100]
In vivo: Male WT mice (Nrf2+/+) and Nrf2-KO (Nrf2−/−) on C57BL/6 Exhaustive incremental treadmill test	SFN pre-training treatment 4 times for 3 days (72, 48, 24, and 3 h) 25 mg/kg bw i.p.	Upregulation of Nrf2 signaling and gene expression of HO-1, NQO1, CAT, and ƴ-GCS in Nrf2+/+ mice skeletal muscle Reduction of TBARS Augmented AMPKα and mtDNA copies Augmented running distance Decreased LDH and CK activities	[101]
Young man performed 6 sets of 5 eccentric exercises with the nondominant arm in elbow flexion 70% MVC	SFN 30 mg/day/2 weeks	Attenuated DOMS and ROM 2 days after exercise Augmented NQO1 mRNA expression in PBMCs Reduction of serum MDA levels 2 days after exercise	[102]
In vivo: male C57BL/6 mice Exhaustive running exercise	SFN 50/mg b/2 h prior to exhaustive running test	Reduction of cytokines TNF-α, IL-1β, and IL-6 Reduction of damage blood biomarkers AST, ALT, and LDH Enhanced mRNA expression of Nrf2 and the downstream enzymes HO-1, SOD1, CAT, and GPx1 in liver tissue	[103]

SFN: sulforaphane; NOQ1: NADPH quinone oxidoreductase 1; HO-1: hem oxygenase 1; Nrf2: E2-factor-related factor; GSH: reduced glutathione; GSSG: oxidized glutathione; CK: creatin kinase; LDH: lactate dehydrogenase; TNFα: factor de necrosis tumoral α; IL-1β: interleukin 1β; NF-κB: nuclear factor kappa B; IκB: inhibitor of kappa B HDAC: histone deacetylase; SMAD7: smad family member 7; PSC: porcine satellite cells; NQO1: NADPH quinone oxidoreductase-1; GST: glutathione S-transferase; GR: glutathione reductase; TAC: total antioxidant activity; CPK: creatin phosphor-kinase; WT: wild type; KO: knock out; mtDNA: mitochondrial DNA; DOMS: delayed onset muscle soreness; ROM: range of motion; MVC: maximum voluntary contraction; PBMCs: peripheral blood mononuclear cells; HO-1: hem oxygenase-1; SOD1: superoxide dismutase 1; CAT: catalase; GPx: glutathione peroxidase.

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
