# Peer review of "Phytochemicals in Skeletal Muscle Health: Effects of Curcumin (from Curcuma longa Linn) and Sulforaphane (from Brassicaceae) on Muscle Function, Recovery and Therapy of Muscle Atrophy"

_plants, 2022, doi:10.3390/plants11192517_

Round 1

Reviewer 1 Report

This review deals with health effect of curcumin and sulforaphane on muscle. Although only two compounds are of interest, they are systematically summarized with reference to the contents of many relevant literatures. I think this manuscript is useful in this field, so it should be published in Plants with minor revision as below:

Line 186

# Curcumin is one of curcuminoid, not flavonoid. Please confirm it.

Figure 1

# The position of “OMe” should be adjusted.

# It should be better to indicate that this structure is enol form.

Line 228

# It should be better to add a figure of chemical structure of “curcumin metabolites”.

Line 420

# What is “NGUG”? Please explain specifically.

Figure 3

# “H3C” in chemical structure is not correct. It should be “H3C”.

# To you can see that C and S are directly bonded, the position of “H3C” should be adjusted.

Author Response

Response to Reviewer 1 Comments

Comments and suggestions for authors:

This review deals with health effect of curcumin and sulforaphane on muscle. Although only two compounds are of interest, they are systematically summarized with reference to the contents of many relevant literatures. I think this manuscript is useful in this field, so it should be published in Plants with minor revision as below:

Point 1: Line 186

# Curcumin is one of curcuminoid, not flavonoid. Please confirm it.

Response 1:  Thank you for the correction, curcumin is curcuminoid and it was changed in the text.

Point 2: Figure 1

# The position of “OMe” should be adjusted.

# It should be better to indicate that this structure is enol form.

Response 2: The chemical structure was modified as you suggested, thank you.

Point 3: Line 228

# It should be better to add a figure of chemical structure of “curcumin metabolites”.

Response 3:  The figure with the chemical structure of “curcumin metabolites” was added.

Point 4: Line 420

# What is “NGUG”? Please explain specifically.

Response 4: “NGUG” refers to Next-Generation Ultrasol Curcumin which is a novel formulation with superior bioavailability compared with natural turmeric extract. This information is provided in the mention reference 63:

Sahin, E.; Orhan, C.; Erten, F.; Er, B.; Acharya, M.; Morde, A.A.; Padigaru, M.; Sahin, K. Next-Generation Ultrasol Curcumin Boosts Muscle Endurance and Reduces Muscle Damage in Treadmill-Exhausted Rats. Antioxidants (Basel, Switzerland) 2021, 10, doi:10.3390/antiox10111692.

Point 5: Figure 3

# “H3C” in chemical structure is not correct. It should be “H3C”.

# To you can see that C and S are directly bonded, the position of “H3C” should be adjusted.

Response 5:  The chemical structure was corrected as you kindly pointed out. Thank you.

Reviewer 2 Report

In this review, the authors summarize the effects of Curcumin curcumin and Sulforaphane on muscle function, recovery, and prevention of muscle atrophy, and analyze their value for muscle health. With the medicinal and food properties, this is a valuable study. However, there are some problems with this manuscript. My comments are as follows.

Main points.

1. The title of this review is formulated by the authors as muscle function as well as muscle atrophy. Muscle includes skeletal muscle, cardiac muscle, and smooth muscle, while the content of this paper is about skeletal muscle. Therefore, the two are not previously equivalent. Suggest revising.

2. This article is about skeletal muscle, but it is not in the keywords, so we suggest adding it.

3. The authors should introduce the severity of skeletal muscle atrophy and the need for treatment in the preamble. And highlight the important role of curcumin and carotenoid in the treatment of skeletal muscle atrophy.

4. The last sentence of the preamble section mentions that curcumin and radiothione can be considered potential targets of nutritional drugs for the treatment of muscle atrophy and recovery, with positive effects on muscle health. And the title is recovery and prevention of muscle atrophy. Prophylactic administration and therapeutic administration are different. Suggested revision.

5. 2.2 mentions the low bioavailability of curcumin and the current application of nanomaterials for prescription design. Relevant treatments can be listed to highlight the cutting edge of research.

6. 2.3.1 talks about muscle disorders. However, there are too many paragraphs in it and the paragraphs are not logical to each other and need to be reorganized. For example, the relevance of the inflammatory response to muscle disorders is highlighted in this section. However, it is not reflected in the third paragraph.

7. the structure design can be divided into points to elaborate on the effect of curcumin and carotenoid pharmacological effects on muscle health.

Secondary points

1. Too many keywords. Three to five are generally recommended. 2.

2. The first sentence of the fifth paragraph in 2.3.1 is not labeled with references.

3. The typography of the literature is misplaced and it is recommended to rearrange it.

4. The full name of the muscle injury indicator CK was not indicated when it was first introduced in the text.

5. the muscle damage marker was introduced as creatine kinase CK in 2.3.2 and as phosphocreatine kinase CPK in 3.3.2. it is suggested to standardize the presentation.

6. The literature summarized by the authors in Table 1 spans a large period of time, and it is recommended to select literature from the last decade.

Author Response

Response to Reviewer 2 Comments

Main points.

Point 1: The title of this review is formulated by the authors as muscle function as well as muscle atrophy. Muscle includes skeletal muscle, cardiac muscle, and smooth muscle, while the content of this paper is about skeletal muscle. Therefore, the two are not previously equivalent. Suggest revising.

Response 1: Thank you for reviewing this manuscript and the comments you made because they help us to improve the quality of this work. The title was adjusted to consider only “skeletal muscle” the same in the content of the manuscript.    

Point 2: This article is about skeletal muscle, but it is not in the keywords, so we suggest adding it. 

Response 2: “skeletal muscle” was added in the keywords.

Point 3: The authors should introduce the severity of skeletal muscle atrophy and the need for treatment in the preamble. And highlight the important role of curcumin and carotenoid in the treatment of skeletal muscle atrophy.

Response 3: The missing information you kindly suggested was added, please see lines 124-131 and 141-143.

Point 4: The last sentence of the preamble section mentions that curcumin and radiothione can be considered potential targets of nutritional drugs for the treatment of muscle atrophy and recovery, with positive effects on muscle health. And the title is recovery and prevention of muscle atrophy. Prophylactic administration and therapeutic administration are different. Suggested revision.

Response 4: The revision was made, and the title is adjusted to “Phytochemicals in Skeletal Muscle Health: Effects of Curcumin (from Curcuma longa Linn) and Sulforaphane (from Brassicaceae) on Muscle Function, Recovery and Therapy of Muscle Atrophy”

Point 5: 2.2 mentions the low bioavailability of curcumin and the current application of nanomaterials for prescription design. Relevant treatments can be listed to highlight the cutting edge of research.

Response 5:  Additional information regarding novel curcumin formulations is described on lines 222-247.

Point 6: 2.3.1 talks about muscle disorders. However, there are too many paragraphs in it and the paragraphs are not logical to each other and need to be reorganized. For example, the relevance of the inflammatory response to muscle disorders is highlighted in this section. However, it is not reflected in the third paragraph.

Response 6:  The information in this section was reorganized as you suggested.

Point 7:  The structure design can be divided into points to elaborate on the effect of curcumin and carotenoid pharmacological effects on muscle health.

Response 7:  Some subtitles were adjusted and added in order to give a better understanding of the information. 

Secondary points.

Point 1: Too many keywords. Three to five are generally recommended. 2.

Response 1: Keywords were changed as you suggest.

Point 2: The first sentence of the fifth paragraph in 2.3.1 is not labeled with references.

Response 2:  The reference was inserted, please see line 317.

Point 3: the typography of the literature is misplaced and it is recommended to rearrange it.

Response 3:   For editing the references, the EndNote X9 program was used as suggested in the preparation of the manuscript in the section “MDPI Reference List and Citations Style Guide” within the Instructions for Authors, it was downloaded the MDPI.ens references style file from the EndNote website at http://endnote.com/downloads/style/mdpi as it is indicated in the section.

Point 4:  The full name of the muscle injury indicator CK was not indicated when it was first introduced in the text.

Response 4: The full name muscle injury indicator “creatine kinase” was added.    

Point 5: the muscle damage marker was introduced as creatine kinase CK in 2.3.2 and as phosphocreatine kinase CPK in 3.3.2. it is suggested to standardize the presentation.

Response 5:  It was used CK to standardize the presentation, thank you.

Point 6: The literature summarized by the authors in Table 1 spans a large period of time, and it is recommended to select literature from the last decade.

Response 6:  Table 1 was modified as you suggested, the oldest literature was removed from the table.

Round 2

Reviewer 2 Report

 Accept

Author Response

Thank you for your comment about English.
Him that before submitting the article to the journal it was reviewed by Maggie Brunner for the English edition. Thank you